



# Hypersaline tidal flats as important "Blue Carbon" systems: A case study from three ecosystems

Dylan R. Brown[1], Humberto Marotta[2,3,4], Roberta B. Peixoto[2,3], Alex Enrich-Prast[2,5,6], Glenda C. Barroso[3], Mario L. G. Soares[7], Wilson Machado[3], Alexander Pérez[3,8], Joseph M. Smoak[9], Luciana M. Sanders[10], Stephen Conrad[1], James Z. Sippo[1,10,11], Isaac R. Santos[1,12], Damien T. Maher[1,10,11], Christian J. Sanders[1,13*]

[1]National Marine Science Centre, School of Environment, Science and Engineering, Southern Cross University, PO Box 4321, Coffs Harbour, NSW 2450, Australia
[2]Ecosystems and Global Change Laboratory (LEMG-UFF) / International Laboratory of Global Change (LINCGlobal), Biomass and Water Management Research Center (NAB), Universidade Federal Fluminense, Av. Edmundo March, s/n°, Niterói, RJ, 24210-310, Brazil
[3]Graduate Program in Geosciences (Environmental Geochemistry), Department of Geochemistry, Universidade Federal Fluminense, Niterói, RJ, 24020-141, Brazil
[4]Physical Geography Laboratory (LAGEF-UFF), Department of Geography, Graduate Program in Geography, Universidade Federal Fluminense (UFF), Av. Gal. Milton Tavares de Souza, s/n°, Niterói, RJ, 24210-346, Brazil.
[5]Department of Thematic Studies–Environmental Change, Linköping University, 581 83, Linköping, Sweden.
[6]Department of Botany, Universidade Federal do Rio de Janeiro, 21941-902, Rio de Janeiro, Brazil.
[7]Laboratory For Mangrove Studies (NEMA-UERJ) / International Laboratory of Global Change (LINCGlobal), Department of Biological Oceanography, Faculty of Oceanography, Universidade do Estado do Rio de Janeiro (UERJ), Rua São Francisco Xavier. 524, sala 4019-E, Rio de Janeiro, RJ, 20550-900, Brazil
[8]Universidad Peruana Cayetano Heredia, Centro de investigación para el desarrollo integral y sostenible (CIDIS), Facultad de Ciencias y Filosofía, Laboratorios de investigación y desarrollo (LID), Laboratorio de Biogeociencias, Av. Honorio Delgado 430, Urb. Ingeniería, Lima 31 – Perú.
[9]Environmental Science, University of South Florida, St. Petersburg, Florida 33701, USA
[10]Southern Cross Geoscience, Southern Cross University, PO Box 157, Lismore NSW 2480, Australia
[11]School of Environment, Science and Engineering, Southern Cross University, PO Box 157, Lismore, NSW, 2480, Australia.
[12]Department of Marine Sciences, University of Gothenburg, Gothenburg, Sweden
[13]State Key Laboratory of Estuarine and Coastal Research and Institute of Eco-Chongming, East China Normal University, Shanghai 201100, P. R. China

*Corresponding author: Christian Sanders (christian.sander@scu.edu.au)

**Abstract.** Hypersaline tidal flats (HTFs) are coastal ecosystems with freshwater deficits often occurring in arid or semi-arid regions near mangrove supratidal zones with no major fluvial contributions. Here, we estimate that organic carbon (OC), total nitrogen (TN) and total phosphorus (TP) are being buried at rates averaging 21 ($\pm$ 6), 1.7 ($\pm$ 0.3), and 1.4 ($\pm$ 0.3) g m$^{-2}$ y$^{-1}$, respectively, during the previous century in three contrasting HTFs systems, one in Brazil (eutrophic) and two in Australia (oligotrophic). Although these rates are lower than those from nearby mangrove, saltmarsh and seagrass systems, the importance of HTFs as sinks for OC, TN and TP may be significant given their extensive coverage. Despite the measured short-term variability between net air-saltpan $CO_2$ influx and emission estimates found during the dry and wet season in the



Brazilian HTF, the only site with seasonal $CO_2$ fluxes measurements, the OC sedimentary profiles over several decades suggests efficient OC burial at all sites. Indeed, the stable isotopes of OC and TN ($\delta^{13}C$ and $\delta^{15}N$) along with C:N ratios show that microphytobenthos are the major source of the buried OC in these HTFs. Our findings highlight a previously unquantified

carbon as well as nutrient sink and suggest that coastal HTF ecosystems could be included in the emerging blue carbon framework.

## 1 Introduction

Hypersaline tidal flats (HTFs), supratidal flats, saltpans, sabkhas, and salt flats are all terms used to define the shallow coastal ecosystems on the upper fringe of fluviomarine plains in estuaries showing freshwater deficits (Ridd and Stieglitz, 2002;

Albuquerque et al., 2014). These environments are generally located in an intermediary position between mangrove forests or saltmarshes and the terrestrial environment which are common in many tropical arid, and to a lesser extent non arid, intertidal zones. These systems occur in many regions around the world including northern Australia, Africa, Spain, the Gulf of Mexico and throughout Brazil where they are referred to as *apicum* ecosystems (Ridd and Stieglitz, 2002; Albuquerque et al., 2013; Albuquerque et al., 2014; Soares et al., 2017). In arid and semi-arid estuaries (Ridd and Stieglitz, 2002) or humid tropical

supratidal zones with less fluvial contribution (Soares et al., 2017), HTF ecosystems cover an area that exceeds mangrove forests and occupy a substantial proportion of tropical intertidal zones. HTFs occupy the area just below the highest astronomical tides and are thus only flooded for short periods of the year (Ridd and Stieglitz, 2002; Bento et al., 2017). Evaporation, the flat topography, and pronounced hydraulic deficit results in hypersaline conditions with salinity as high as five times that of seawater (Ridd and Stieglitz, 2002; Shen et al., 2018).

60           Despite the extreme conditions and the apparent absence of vegetation, microphytobenthos are commonly found on the surface of HTFs (usually in the form of microbial mats dominated by cyanobacteria from the Oscillatoriales order including *Microcoleus* spp., *Leptolyngbya* spp. and *Lyngbya* sp.) (Adame et al., 2012; Masuda and Enrich-Prast, 2016). These microphytobenthos are well adapted to the extreme conditions (Paerl et al., 2000) and are considered to be the main primary producers in HTFs. Similar to traditional vegetated blue carbon systems (Ouyang and Lee, 2014; Sanders et al., 2016a;

Macreadie et al., 2019; Serrano et al., 2019), these microphytobenthos and are capable of high rates of carbon (C) and nitrogen (N) fixation from the atmosphere, particularly after periods of flooding and/or rainfall (Chairi et al., 2010; Adame et al., 2012; Burford et al., 2016). Their ability to sequester and potentially store carbon and nutrients in their soils for long periods of time (centuries to millennia) makes them noteworthy contenders to be included in blue carbon framework (Lovelock and Duarte, 2019). Upon inundation, the fixed C, N and other nutrients such as phosphorus (P) may be leached from the microbial mats

and transported to adjacent coastal areas, where nutrient subsidies can enhance the overall productivity of the receiving ecosystems (Adame et al., 2012; Burford et al., 2016).

           Given the few studies on HTFs, there is limited understanding on the role that these ecosystems play in the coastal zone, and whether they are currently under threat from global change (Halpern et al., 2008; Martinez-Porchas and Martinez-



Cordova, 2012). To date, there has been large scale destruction and degradation of these systems on a global scale as a result

of anthropogenic pressures on coastal areas including infilling for urban and agricultural/aquaculture development (Halpern et al., 2008). Although there has been the implementation of various laws in some parts of the world to prevent the loss of coastal vegetated systems, these legislations rarely extent to protect HTFs that are viewed as being ecological deserts with no obvious vegetation (Albuquerque et al., 2013). Furthermore, the landward encroachment of mangrove forests as a response to rising sea levels, coupled to barriers preventing landward migration of HTF (i.e. the "coastal squeeze"), may also contribute to the

loss of these ecosystems (Alongi, 2008; Saintilan et al., 2014; Kelleway et al., 2017).

Given the substantial areal extent of these HTFs and the fact that they remain relatively undisturbed in many regions around the world, HTFs may have unrecognized ecological values (Burford et al., 2016). However, information on OC, nitrogen and phosphorus burial and sediment $CO_2$ fluxes from these ecosystems remains scarce (e.g. Bento et al., 2017; Schile et al., 2017). Determining if HTFs are a source or sink of carbon is critical to understanding their importance and value in

regards to climate change and coastal carbon sequestration (Lovelock and Duarte, 2019). Here, we quantify carbon and nutrient burial and atmospheric $CO_2$ fluxes in HTFs in Australia and Brazil. We hypothesize that microphytobenthos in HTFs sequester $CO_2$ from the atmosphere and a portion of this organic matter (and associated nitrogen and phosphorus) is buried, similar to the traditional vegetated blue carbon systems.

## 2 Methods

### 2.1 Study site

This study was conducted in three tropical HTFs in Australia and Brazil (Fig. 1). In Australia, the HTF study sites were located near Karumba, Queensland (17 º25'12"S, 140º51'36"E) and Curtis Island, Gladstone, Queensland (Site 1: 23º45'41"S, 151º16'34"E; Site 2: 23º45'18"S, 151 º16'49"E), and in Brazil the study site was located in Guaratiba, Rio de Janeiro (23º00'29"S, 43º36'31"W).

In Australia, the Karumba HTF is located adjacent to the oligotrophic mouth of the Norman River estuary in the southeastern coast of the Gulf of Carpentaria. The study site consists of a large continuous HTF (16.9 km$^2$) in the high upper intertidal zone. The southeastern Gulf of Carpentaria has a diurnal tidal cycle (typical range <0.1 – 4.5 m) and a 23-year average annual rainfall of 833 mm (based on monthly averages from 1938 to 2010) with most falling in the summer monsoon period (790 mm from December to March) (Bureau of Meteorology, 2019). The 19-year average maximum monthly

temperatures (from 1993 to 2019) vary from 27 ºC in the dry winter months to 33 ºC in the wet summer months (Bureau of Meteorology 2019). A narrow strip of mangrove forest followed by extensive tidal mudflats fringe the HTFs on the seaward side.

The Gladstone Harbour experiences similar tidal and climatic conditions to Karumba with semidiurnal tides (typical range 0.1 – 4.7 m) and a 25-year average annual rainfall of 846 mm (based on monthly averages from 1994 to 2019); with

most falling also in the summer monsoon period (537 mm from December to March) (Bureau of Meteorology, 2019). The 26-





year average maximum monthly temperatures (from 1993 to 2019) vary from 23 ℃ in the winter months to 31 ℃ in the summer months (Bureau of Meteorology, 2019). The sheltered strait between Curtis Island and the mainland of Australia is largely occupied by mangrove forests and large continuous expanses of HTFs. The Gladstone site contained two HTF study areas; Site 1 (2.84 km$^2$) was situated in the higher tidal area and is inundated less frequently and for shorter periods of time than Site 2 (0.95 km$^2$).


In Brazil, the tropical HTF was located in the Guaratiba State Biological Reserve, ~40 km south of Rio de Janeiro City which forms part of the Sepetiba Bay estuary system. This conservation area is surrounded by the urban expansion area of Rio de Janeiro City, and Sepetiba Bay receives discharges of nutrients and organic matter from its watershed dominated by agriculture, pasture and urban uses (Rezende et al., 2010). The HTF covers an area of approximately 7.4 km$^2$ equivalent to almost 36% of the fringing mangrove forest (Estrada et al., 2013; Soares et al., 2017) (Table 1). There is little variation in


topography and the tidal range is 0.1 – 2.0 m (Masuda and Enrich-Prast, 2016; Bento et al., 2017). The 32-year monthly average rainfall and temperature vary from 36 mm and 21 ℃ in dry winter months and 138 mm and 27 ℃ in rainy summer months, reaching annual averages of accumulated rainfall of 1058 mm (Estevam, 2019) (Table 1).

## 2.2 Sediment core sampling and analysis


Sediment cores (one core per site for a total of four cores) were collected from the middle of HTFs by using either a 50 cm long, 5 cm diameter Russian Peat Auger (Karumba core) or by inserting a PVC tube (8.7 cm diameter) into the substratum using manual percussion (Gladstone and Guaratiba cores). Only cores with no observed compaction were retained for further analysis. The sediment cores were sectioned at 1 cm intervals (with the exception of the Karumba core which was sectioned at 2 cm intervals). Dry bulk density (DBD, g cm$^{-3}$) was determined as the dry sediment weight (g) divided by the initial volume


(cm$^3$) (Ravichandran et al., 1995). From the original dry section, a non-homogenized portion was rewetted and treated with 30% hydrogen peroxide (H$_2$O$_2$) to remove organic matter without altering grainsize. A solution of sodium hexametaphosphate was used as a deflocculating agent to separate aggregates prior to grain size analysis. Grain size analyses were conducted using a CILAS 1090L diffraction laser unit or wet sieving following the methods used by Conrad et al., (2019). Total phosphorous (TP) was measured after acid digestion (H$_2$O/HF/HClO$_4$/HNO$_3$, 2:2:1:1) using a Perkin Elmer ELAN DRCe ICPMS.


Organic carbon (OC) and total nitrogen (TN) stable isotope ratios of mangrove leaves, microphytobenthos, and HTF sediments were measured to identify the sources of organic matter (OM) contributing the sediment column at each site. Fresh green leaves from mangrove trees (n = 3 for each dominant species: *Rhizophora mangle*, *Avicenna shaueriana*, and *Laguncularia racemose*) were collected at 1-2m above the soil, and washed with deionized water soon after sampling in the Brazilian HTF. Samples were then lyophilized, crushed, sieved, and ~6-8 mg encapsulated in tin capsules to determine the


OC, TN and their isotopic composition (δ$^{13}$C and δ$^{15}$N). Microphytobenthos samples, in the form of dense algal mats, were collected from the surface of HTF sediments, scrapped and thoroughly washed with deionized water to avoid sediment contamination. A total 6 microphytobenthos samples were collected and analyzed (3 from Brazil, 2 from Karumba, and 1 from Gladstone). A homogenized portion was acidified to remove carbonate material, washed in deionized water, dried (60 °C) and





then ground to powder for OC and $\delta^{13}C$ analyses using a Leco Flash Elemental Analyzer coupled to a Thermo Fisher Delta V

IRMS (isotope ratio mass spectrometer). A non-acidified homogenized portion was also analyzed for TN and $\delta^{15}N$. Analytical precision: C = 0.1%, N = 0.1%, $\delta^{13}C$ = 0.1‰ and $\delta^{15}N$ = 0.15 ‰. We assess whether HTFs accumulate carbon and then to compare HTF with well-established, nearby mangrove systems.

Radionuclides from the uranium-238 ($^{238}U$) decay series were measured in a high-purity germanium (HPGe) planar or well gamma detectors. Identical geometry was used for all samples and sample dry weights were between 20 and 30 g.

Sealed and packed samples were set aside for at least 21 days to allow for radon-222 ($^{222}Rn$) ingrowth and to establish secular equilibrium between radium-226 ($^{226}Ra$) and its granddaughter lead-214 ($^{214}Pb$). Lead-210 ($^{210}Pb$) activity was determined by the direct measurement of the 46.5 KeV gamma peak. $^{226}Ra$ activity was determined via the $^{214}Pb$ daughter at 351.9 KeV. $^{210}Pb$ and $^{226}Ra$ activities were calculated by multiplying the counts per minute by a correction factor that includes the gamma-ray intensity and detector efficiency determined from standard calibrations. Excess $^{210}Pb$ was used to determine ages of sediment

intervals using the Constant Initial Concentration (CIC) model (Appleby and Oldfield, 1992). Mass accumulation rates were multiplied by the percent OC, N and TP to calculate burial rates.

### 2.3 Air-sediment gas flux measurements

$CO_2$ fluxes at the air-sediment interface were measured in July 2009 & 2010 and February 2015 (Guaratiba, Brazil), August 2016 & 2018 (Karumba, Australia), and June 2018 (Gladstone, Australia), encompassing the annual variation of emissions

between dry and rainy seasons in the HTF in Brazil and non-monsoon months in Australia. In all sampling sites, we used sediment chambers connected in a closed system with an infrared or cavity ring-down analyzer as reported in Lovelock (2008). The sediment chambers were composed of transparent plexiglass (light chamber) or an opaque material such as PVC or covered by layers of aluminium foil (dark chamber) for measurements of light and dark air-sediment $CO_2$ fluxes respectively (Leopold et al., 2015). Before each measurement, the chambers were gently pushed into the sediment (~2 cm) to form a gas tight seal.

Each short-term incubation lasted 5-15 min to achieve a linear change in $CO_2$ concentration within the chambers. Gas concentrations were measured using either a Los Gatos Research (LGR) Ultra-Portable Greenhouse Gas Analyzer (UGGA) or Picarro G4301 GasScouter recorded at 1-second intervals in the Australian sites, and a PPSystems EGM-4 or a Vaisala GMT222 at 1-minute intervals in the Brazilian sites. Equipment were previously calibrated with $CO_2$ standards of 400 and 1000 ppm in the laboratory.

$CO_2$ fluxes were measured at dark and light conditions in Brazil (n = 51 and 94, respectively) and Australian HTFs (n = 46 and 32, respectively). The air-sediment $CO_2$ fluxes were calculated from the maximum linear change in $CO_2$ concentration over the duration of the measurement using the following formula (Rosentreter et al., 2017 and references therein):

$$F = (s(V/RT\text{air}))A \qquad\qquad (1)$$

where $s$ is the regression slope for each chamber incubation deployments (ppm sec$^{-1}$ or ppm min$^{-1}$, converted to ppm h$^{-1}$), $V$ is the chamber volume (m$^3$), $R$ is the universal gas constant, $T_{air}$ is the air temperature inside the chamber (K), $A$ is the surface



area of sediment inside the chamber (m$^2$). Negative values represent net sediment $CO_2$ uptake while those positive represent net $CO_2$ emission from sediments to the atmosphere. We assume that pressure in the chamber is 1 atm. To determine the net ecosystem exchange (NEE), we integrate diurnal and night fluxes from light and dark chambers for each sampling day,

respectively. To test the normality of $CO_2$ emissions data we performed a Komogorov-Smirnov test. For non-normally distributed data, a Mann Whitney test (significance level; $p < 0.05$) was undertaken to compare light and dark fluxes at the combined Brazil and Australian samples, and also to compare wet and dry season Brazil fluxes.

## 3 Results

### 3.1 Sediment accretion rates (SAR)

All four sediment profiles showed a net down-core decrease in excess $^{210}$Pb activity reaching background levels at the bottom of each sediment core (Fig. 2), enabling the use of the CIC $^{210}$Pb dating methodology. All cores were dated back to between 50 and 110 years with constant sediment accretion rates estimated at $0.11 \pm 0.05$ (1903), $0.18 \pm 0.06$ (1955), $0.21 \pm 0.05$ (1931), and $0.23 \pm 0.05$ cm yr$^{-1}$ (1964) for Guaratiba, Karumba, Gladstone site 1, and Gladstone site 2 HTF sediment cores, respectively.

### 3.2 Carbon, nitrogen and phosphorus burial rate estimates

Most of the parameters remained relatively constant throughout the sediment profiles, with no clear vertical trends in grain size, OC, TN or TP (Fig. 3). Sand content was generally <20% and OC, TN and TP contents ranged from 0.09 to 1.40, 0.01 to 0.16, and 0.02 to 0.12%, respectively across all sites and depth intervals (Fig. 3). By multiplying the average sedimentation rate, DBD, and OC content in these cores, we obtained carbon burial rates of 17.8 ($\pm$ 0.8), 31.7 ($\pm$ 4.3), 11.3 ($\pm$ 2.1), and 25.2

($\pm$ 2.9) g m$^{-2}$ y$^{-1}$ in the Guaratiba, Karumba, Gladstone site 1, and Gladstone site 2 cores respectively for the past ~50 years (Table 2). Average TN burial rates were 2.3 ($\pm$ 0.2), 2.8 ($\pm$ 0.3), 0.8 ($\pm$ 0.1), and 1.2 ($\pm$ 0.1) g m$^{-2}$ y$^{-1}$ and average TP burial rates were 2.0 ($\pm$ 0.1), 1.3 ($\pm$ 0.1), 1.4 ($\pm$ 0.0), and 1.4 ($\pm$ 0.3) g m$^{-2}$ y$^{-1}$ in the Guaratiba, Karumba, Gladstone site 1, and Gladstone site 2 cores, respectively (Table 2).

### 3.3 Organic matter source

To assess the source of organic matter (OM), sediment, HTF microphytobenthos, and nearby mangrove end member samples were analysed for $\delta^{13}$C stable isotopes and cross-plotted against molar C:N ratios (Fig. 4). Microphytobenthos samples showed a small spread in $\delta^{13}$C and molar C:N ratios ranging from -13.4 to -19.0 ‰ and 7.9 to 14.8 respectively (Fig. 4). Similarly, values of $\delta^{13}$C and molar C:N ratios showed little down-core variation in both the Guaratiba (-17.7 to -18.4 ‰ and 7.6 to 9.7 respectively) and Karumba sediment cores (-15.5 to -20.5 ‰ and 10.5 to 14.6 respectively). In contrast, both the Gladstone

sediment cores showed a considerable range in the $\delta^{13}$C and molar C:N values (-20.1 to -24.2 ‰ and 13.6 to 21.8 at site 1; -16.6 to -24.4 ‰ and 10.7 to 34.8 at site 2). Higher $\delta^{15}$N and lower C:N ratio values were noted in the Guaratiba HTF compared to other sites (Fig. 4).

### 3.4 $CO_2$ fluxes at the air-sediment interface



Median ($\pm$ SE) hourly $CO_2$ fluxes measured at the air-sediment interface varied with HTF location and type of measurement

(Light vs Dark) (Fig. 5). Median light $CO_2$ values were -2.1 ($\pm$ 4.1), 38.2 ($\pm$ 0.0), 13.7 ($\pm$ 1.9), and 29.3 ($\pm$ 2.1) mg C $m^{-2}$ $h^{-1}$

for the Brazilian (Guaratiba) and Australian (Karumba, Gladstone site 1, and Gladstone site 2) HTFs, respectively (Fig. 5).

Median $CO_2$ fluxes in the dark chambers were significantly higher than those estimated in the light chambers (Mann Whitney

test; $p<0.05$), *i.e.* 2.1 ($\pm$ 1.0), 39.6 ($\pm$ 9.2), 45.7 ($\pm$ 4.5), and 34.6 ($\pm$ 3.1) mg C $m^{-2}$ $h^{-1}$ for Guaratiba, Karumba, Gladstone site

1, and Gladstone site 2, respectively (Fig. 5). In Brazil, significantly higher $CO_2$ uptake rates (median $\pm$ SE) were recorded in

the light chambers during the dry season compared to the wet season (-3.0 $\pm$ 1.3 and 48.9 $\pm$ 7.2 mg C $m^{-2}$ $h^{-1}$, respectively;

Mann Whitney; $p<0.05$).

## 4 Discussion

### 4.1 C, N and P burial in HTFs versus vegetated blue carbon ecosystems

Considerable differences in OC burial rates between the two Gladstone sites were observed in this study. The likely difference

between sites is due to the tidal area of each site, i.e. upper vs lower tidal areas are expected to accumulate carbon at different

rates (Sanders et al., 2014). By averaging the sediment burial rates on a centennial scale (*i.e.* entire core) of the four sediment

cores across all the study sites, we estimate that HTF ecosystems accumulate OC, TN and TP at rates of 21 ($\pm$ 6), 1.7 ($\pm$ 0.3),

and 1.4 ($\pm$ 0.3) g $m^{-2}$ $y^{-1}$, respectively. These centennial scale averages reduce short term variations allowing comparisons with

saltmarsh, mangrove forests, and seagrass beds which have been studied extensively using similar methodologies and

timeframes (McLeod et al., 2011). The average OC accumulation rates in HTF systems were ~12, ~8 and ~7 fold lower than

the global averages reported for saltmarsh (245 $\pm$ 26 g $m^{-2}$ $y^{-1}$; Ouyang and Lee (2014)), mangrove forests (163 $\pm$ 40 g $m^{-2}$ $y^{-1}$;

Breithaupt et al. (2012)), and seagrasses (138 $\pm$ 38 g $m^{-2}$ $y^{-1}$; McLeod et al. (2011)), respectively. These lower burial rates may

be related to the lower organic matter supply (including no contribution from below ground productivity) and/or lower

sediment accretion rates than the traditional blue carbon systems. Furthermore, the reduced structural complexity and ability

of the microalgae to trap sediments, the lower primary production rates, the lack of underground root protection, and the fact

that microalgae organic material is more labile can explain the lower burial and sediment accretion rates of HTFs than

traditional, vegetated blue carbon systems.

        Hypersaline tidal flats can be a significant source of nutrient export to adjacent ecosystems which may potentially

fuel primary productivity in nutrient-limited receiving marine ecosystems (Lovelock et al., 2010; Burford et al., 2016). Here,

we find that these HTF ecosystems are also sites for the long-term storage of nitrogen and phosphorus (Table 2). The high TP

burial rates compared to TN observed are likely due to the lack of anthropogenic nitrogen inputs observed in other systems.

Although the average TN accumulation rates reported here (1.7 $\pm$ 0.3 g $m^{-2}$ $y^{-1}$) were also relatively low when compared to

mangrove sediments (12.5 $\pm$ 1.9 g $m^{-2}$ $y^{-1}$; Breithaupt et al. (2014)), the average TP accumulation rates in both Australian

pristine (1.4 $\pm$ 0.3 g $m^{-2}$ $y^{-1}$) and Brazilian eutrophic HTFs (2.0 $\pm$ 0.1 g $m^{-2}$ $y^{-1}$) were higher than conserved mangrove sites with

little anthropogenic nutrient discharges (0.5 $\pm$ 0.2 g $m^{-2}$ $y^{-1}$; Breithaupt et al. (2014)). However, the HTF TP accumulation rates

were not as high as those found in anthropogenically disturbed mangrove sites such as the heavily urbanized Jiulongjiang



Estuary, China with TP accumulation rates reaching 48.1 g m$^{-2}$ y$^{-1}$ (Alongi et al., 2005). Anthropogenic activities such as urbanization and major industrial developments drive degradation and increased primary production in mangrove forests (Sanders et al., 2014). Nutrients such as iron and phosphorus may be limiting to mangrove growth (Alongi, 2010; Reef et al.,

2010), and those forests receiving high nutrient loads from highly concentrated  anthropogenic nutrient discharges accumulate OC, TN, and TP at rates much higher than those from the undisturbed mangrove (Sanders et al., 2014).  Nevertheless, the nitrogen and phosphorus burial in HTFs as shown here over long periods of time may play an important role in nutrient sequestration from other coastal anthropogenic activities, e.g. shrimp farming activities (Ashton, 2008; Marchand et al., 2011).

By upscaling the average OC, TN and TP accumulation results for the past century in this study to the regional areas
of HTFs, we can provide a first-order estimate of the amount of OC, TN and TP being stored annually in these HTFs. Ridd and Stieglitz (2002) identify the areal extent of both HTFs and mangrove forests for five estuaries in Queensland, Australia with the HTFs identified as having a ~10-fold higher areal extent (279 km$^2$) than mangrove forests (29 km$^2$) over the five estuaries. In these estuaries alone, HTFs would contribute to the annual accumulation of approximately 5.76 ± 1.57, 0.46 ± 0.09 and 0.40 ± 0.08 Gg y$^{-1}$ of OC, TN and TP respectively which is similar to the contribution of mangrove forests (4.73 ±
1.16, 0.36 ± 0.06 and 0.26 ± 0.03 Gg y$^{-1}$ for OC, TN and TP respectively) when based on global average accumulation rates (Breithaupt et al., 2012; Breithaupt et al., 2014). In contrast to Australia, the mangrove forests of Guaratiba (20.9 km$^2$) have been identified to have a ~3 fold higher area than local HTFs (7.4 km$^2$) (Soares et al., 2017); resulting in annual OC, TN and TP accumulation in HTFs (0.15 ± 0.04, 0.01 ± 0.00 and 0.01 ± 0.00 Gg y$^{-1}$ respectively) equivalent to 4-6% of those estimated for mangrove forests (3.41 ± 0.84, 0.26 ± 0.04 and 0.19 ± 0.02 Gg y$^{-1}$ for OC, TN and TP respectively) when based on global
average accumulation rates (Breithaupt et al., 2012; Breithaupt et al., 2014).

Our estimates suggest HTFs are capable of long-term storage of OC, TN and TP and given their large areal extent, have the potential to store as much OC, TN and TP as traditional coastal blue carbon systems in arid regions such as Queensland, Australia. To improve these estimates, there is clearly a need for determining carbon and nutrient accumulation rates from additional coastal HTFs and assess their areal cover in Australia, Brazil and elsewhere.

**4.2 Organic matter source**

Microphytobenthos associated with coastal HTF ecosystems were an important source of OM accumulation in each of the sediment profiles. Microscopic examinations in previous studies have identified the cyanobacteria *Oscillatoria* spp., *Lyngbya* spp., *Microcoleus* spp., and *Phormidium* spp. as the dominant microphytobenthos in HTF ecosystems (Adame et al., 2012; Burford et al., 2016; Masuda and Enrich-Prast, 2016; Bento et al., 2017). These microphytobenthos are likely to be the
important species contributing to the accumulation of OM, particularly in Guaratiba and Karumba where the δ$^{13}$C and C:N ratio values were consistently similar to that of the HTF microphytobenthos end member values (Fig. 4). Therefore, we suggest that microphytobenthos are the dominate source of OM accumulating in the sedimentary substrates during the past century.

In contrast to the Guaratiba and Karumba profiles, the considerable spread in δ$^{13}$C and molar C:N ratio values along the Gladstone sedimentary profiles suggests the OM accumulation inputs are from a combination of microphytobenthos and
mangrove material (Fig. 4). These results are not surprising given the vast areal extent of mangrove systems in the Gladstone





Harbour and their close proximity to the HTFs. Effective N consumption in coastal wetland sediments (Wadnerkar et al., 2019) may increase overall sedimentary C:N ratios. Sedimentary N and the relatively higher $\delta^{15}N$ values observed in the Guaratiba HTF sediments (Fig. 3) may be indicative of eutrophication (Sanders et al., 2014). Indeed, wastewater inputs typically have elevated $\delta^{15}N$ values due to elevated nitrogen cycling including denitrification (Costanzo et al., 2005). Anthropogenic

wastewater inputs high in N and P loads are also of growing concern across the globe, particularly in HTF areas near shrimp farming (Ashton, 2008; Marchand et al., 2011). While there are no shrimp farms near our study sites, the release of high N and P loads may drive eutrophication of adjacent coastal areas (Ashton, 2008; Marchand et al., 2011) and modify carbon burial rates (Sanders et al., 2014). In addition to the increase of the N and P release, shrimp farms would drive a reduction of the HTFs area that may remove N and P.

**4.3 $CO_2$ fluxes at the air-sediment interface**

The great variability of air-saltpan $CO_2$ fluxes here suggests a highly dynamic and productive metabolism along the HTFs. The oligotrophic Gladstone's sites were net sources of $CO_2$ to the atmosphere in the dry season ($0.72 \pm 0.01$ g C m$^{-2}$ d$^{-1}$), while the eutrophic Guaratiba's HTF experienced net $CO_2$ uptake and source in the dry and rainy season ($-0.03 \pm 0.01$ and $0.71 \pm 0.22$ g C m$^{-2}$ d$^{-1}$, respectively). These estimates of net seasonal fluxes of $CO_2$ contribute to reduce the scarcity of studies

quantifying this gas exchange at the air-sediment interface in HTFs (Table 3). The net $CO_2$ source observed during rainy compared to the net influx during dry seasons in Brazil (Mann-Whitney, $p<0.05$) may be attributed to higher temperature and cloud cover over sampling days in the rainy summer than the dry winter. Previous evidence indicates that the light attenuation by clouds may reduce microphytobenthos photosynthetic activity (Blackford, 2002; Barnett et al., 2020), while warmer sampling conditions on average $\pm$ SE, $26.7 \pm 0.02$ and $21.5 \pm 0.02°C$ during the wet and dry season, respectively, may stimulate

heterotrophy in tidal flat systems (Laviale et al., 2015; Lin et al., 2020). The $CO_2$ source to the atmosphere found during the rainy summer still contrasted with previous evidence in the same Brazilian HTF on enhanced $CO_2$ sink after rain events in winter (Bento et al., 2017), suggesting that factors other than the occurrence of precipitation (*e.g.*, rainfall duration and intensity) may cause the dynamic short-term changes on microphytobenthic production. In addition, higher values of air-saltpan $CO_2$ influx during similar sunnier periods in the Brazilian than Australian HTFs may be attributed to more eutrophic

conditions, which could stimulate microphytobenthic production in saltpan sediments (Xie et al., 2019). These findings highlight the high temporal variability and the need for future seasonal sampling due to the short-term shifts in air-saltpan $CO_2$ exchange, specifically considering the potential net atmospheric $CO_2$ sink in HTFs as indicated by the autochthonous OM in found in the sedimentary profiles. As such, gaining a clearer understanding of the drivers of net primary production in HTFs during changing climatic and anthropogenic conditions is critical to determine their global relevance as atmospheric carbon

sinks.

**4.4 Can HTFs be considered "Blue Carbon" systems?**

While much of the research on blue carbon systems continues to focus on mangrove forests, tidal marshes, and seagrass meadows, there are suggestions of considering other ecosystems in the blue carbon framework (Raven, 2018; Trevathan-Tackett et al., 2015; Lovelock and Duarte, 2019). Tidally influenced freshwater forests, marine macroalgae and kelp beds, and





HTFs for instance, are all ecosystems where blue carbon stocks and sequestration rates may be conceptually equivalent to conventional blue carbon systems (Raven, 2018; Krause-Jensen et al., 2018; Krauss et al., 2018; Lovelock and Duarte, 2019).

Lovelock and Duarte (2019) discuss several key assessment criteria for the inclusion of an ecosystem in the blue carbon framework. First, an ecosystem needs to be capable of long term-storage of $CO_2$ resulting in significant greenhouse gas (GHG) removal from the atmosphere. The results from this study indicate that HTF ecosystems are capable of long-term-

storage of fixed $CO_2$ at rates averaging $21 \pm 6$ g C m$^{-2}$ y$^{-1}$. Given that HTFs are extensively distributed in coastal areas showing freshwater deficit such as in northern Australia and Brazil, the scale of $CO_2$ removal can be significant and comparable to traditional blue carbon systems in some key arid regions. While this study demonstrates carbon burial in three HTF systems, accurate estimates on the magnitude of this carbon sink on national or global scales will require further studies and improved areal estimates.

The second consideration for inclusion into the blue carbon framework is that management of an ecosystem is possible. Management should maintain or enhance carbon and nitrogen stocks and thereby reduce GHG emissions (Lovelock and Duarte, 2019). Over the past few decades, HTFs have experienced large scale destruction and degradation on a global scale as a result of anthropogenic pressures such as urban and agricultural/aquaculture development (Ashton, 2008; Halpern et al., 2008; Martinez-Porchas and Martinez-Cordova, 2012) which may ultimately lead to large scale release of $CO_2$ to the

atmosphere. Local, national and/or international management actions, therefore, have the potential to reduce and possibly revert these losses and destruction, thereby maintaining or even enhancing C sequestration similar to adjacent mangroves and saltmarshes. These management practices include regulating urban development or the construction of shrimp farming to prevent HTF ecosystem decline (Halpern et al., 2008; Martinez-Porchas and Martinez-Cordova, 2012). Moreover, current frameworks and management strategies in place for coastal vegetated ecosystems have the potential to incorporate HTFs given

their close association. Therefore, we suggest that HTF ecosystems can be classified as blue carbon systems and should be included in global management and mitigation polices and are likely to be important contributors on regional scales.

## 5 Conclusions

The investigated HTF ecosystems have accumulated significant amounts of OC, TN and TP during the previous century. Although these accumulation rates are lower than other vegetated blue carbon systems per unit area, a substantial amount of

carbon and nutrients are sequestered in HTFs considering their extensive global areal extent and should not be overlooked. Stable isotope analysis along with the molar C:N ratios indicate that the microphytobenthos associated with these HTFs are an important source of the organic material accumulated along the sediment columns of these systems. To improve the robustness of our observations, there is a need for determining carbon and nutrient accumulation rates and $CO_2$ fluxes from additional coastal HTFs and to determine a more precise areal estimate of HTFs in Australia, Brazil and other parts of the world. However,

our initial data implies that these coastal HTF ecosystems fit the definition of blue carbon systems and could be included in global and regional management and mitigation polices.



## Acknowledgments

Field and laboratory investigations were funded by the Australian Research Council (DE160100443, DP180101285 & LE140100083). The data used in this research are available in the Tables. HM and RBP were funded by Coordenação de

Aperfeiçoamento de Pessoal de Nível Superior (CAPES - Código 001). HM was awarded by CNPq Research Productivity and FAPERJ Young Scientist of Rio de Janeiro State fellowships. AP is supported by the "Fondo Nacional de Desarrollo Científico Tecnológico y de Innovación Tecnológica" (FONDECYT - PERU) thought the Magnet program (Grant n° 007-2017-FONDECYT) and the "Incorporación de Investigadores" program (Grant n° E038-2019-02-FONDECYT-BM).

## Conflict of Interest

The authors declare that the research was conducted in the absence of any commercial or financial relationships that could be construed as a potential conflict of interest.

## Author Contributions

DRB, HRM, and CJS designed and obtained funding for this work. DRB, CJS, HRM, RBP, DTM and LSM contributed to acquisition of data, and contributed to the analysis and interpretation of data. All of the authors made contributions to the

drafting of the article and revisions critically for important intellectual content. All authors gave the final approval of the version to be published.

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

**Table 1. Characterization of study sites. Rainfall (mm) and Temperature (ºC) data are based on annual averages derived from**
**monthly measurements (n = number of years data) and HTF: Mangrove Area indicates the ratio of hypersaline tidal flat area to mangrove area.**

| | Hypersaline Tidal Flat Study Sites | | | |
| --- | --- | --- | --- | --- |
| | Guaratiba | Karumba | Gladstone Site 1 High Tidal Area | Gladstone Site 2 Low Tidal Area |
| Location | 23º00'29"S, 43º36'31"W | 17º25'12"S, 140º51'36"E | 23º45'41"S, 151º16'34"E | 23º45'18"S, 151 º16'49"E |
| Rainfall (mm) | 1058 (n=32) | 883 (n=23) | 846 (n=25) | 846 (n=25) |
| Temperature (ºC) | 21 – 27 (n=32) | 27 – 33 (n=19) | 23 – 31 (n=26) | 23 – 31 (n=26) |





| | | | | |
|---|---|---|---|---|
| Tidal Range (m) | 0.1 – 2.0 | 0.1 – 4.5 | 0.1 – 4.7 | 0.1 – 4.7 |
| HTF Area (Km$^2$) | 7.4 | 16.9 | 2.84 | 0.95 |
| HTF: Mangrove Area | 1:3 | 10:1 | 1:1 | 1:1 |

**Table 2. Mean (± standard error) ~50 year organic carbon (OC), total nitrogen (TN) and total phosphorus (TP) burial rates in the four hypersaline tidal flat sediment cores. Means are based on one core per site**

| Study site | OC (g m$^{-2}$ y$^{-1}$) | TN (g m$^{-2}$ y$^{-1}$) | TP (g m$^{-2}$ y$^{-1}$) |
|---|---|---|---|
| Guaratiba | 17.8 ± 0.8 | 2.3 ± 0.2 | 2.0 ± 0.1 |
| Karumba | 31.7 ± 4.3 | 2.8 ± 0.3 | 1.3 ± 0.1 |
| Gladstone site 1 | 11.3 ± 2.1 | 0.8 ± 0.1 | 1.4 ± 0.0 |
| Gladstone site 2 | 25.2 ± 2.9 | 1.2 ± 0.1 | 1.4 ± 0.3 |






**Table 3. Mean sediment organic carbon content (%) and CO$_2$ fluxes (mg C m$^{-2}$ h$^{-1}$) at the air-sediment interface in hypersaline tidal flat sediments reported in the literature. Values are means ± standard error unless otherwise stated.**

| Location | Air-sediment CO$_2$ fluxes (light) (mg C m$^{-2}$ h$^{-1}$) | Air-sediment CO$_2$ fluxes (dark) (mg C m$^{-2}$ h$^{-1}$) | Sediment organic carbon content (%) | Reference |
|---|---|---|---|---|
| New Caledonia | 20.4 ± 3.2 | 20.0 ± 3.3 | 1.6 ± 0.2 [d] | Leopold et al. (2013) |
| Guaratiba, Brazil | | | | |
| Dry season | -6.7 ± 1.3 | 2.3 ± 0.3 | 0.6 ± 0.0 | This study |
| Wet season | 44.2 ± 7.2 | 20.9 ± 3.7 | 0.6 ± 0.0 | This study |
| Guaratiba, Brazil | -5.3 ± 3.2 [b] | 4.8 ± 3.6 [b] | NA | Bento et al. (2017) |
| Gladstone, Australia | 24.3 ± 2.0 | 36.5 ± 2.7 | 0.5 ± 0.1 | This study |
| Karumba, Australia | 38.2 ± 0.0 | 44.8 ± 9.2 | 0.8 ± 0.1 | This study |
| Karumba, Australia | 14.5 ± 12.2 [a,b] | 0.9 ± 0.5 [a,b] | NA | Burford et al. (2016) |
| Exmouth Gulf, Australia | 64.8 ± 7.4 [a] | 40.2 ± 3.7 [a] | 0.7 | Lovelock et al. (2010) |
| Arabian Gulf, United Arab Emirates | 38.0 ± 15.1 [b,e] | | 1.6 ± 0.7 [b] | Schile et al. (2017) |
| Murcia, Spain | NA | NA | 5.2 ± 0.5 | Conesa et al. (2011) |
| Tunisia, Africa | NA | NA | 0.6 ± 0.1 | Chairi et al. (2010) |
| Ceará, Brazil | NA | NA | 0.5 ± 0.3 | Albuquerque et al. (2014) |
| Ceará, Brazil | NA | NA | 0.7 ± 0.8 | Albuquerque et al. (2013) |
| Bahia, Brazil | NA | NA | 0.7 ± 0.1 [c] | Albuquerque et al. (2013) |
| Teremba Bay, New Caledonia | NA | NA | 5.9 ± 1.2 [b] | Marchand et al. (2011) |
| Tempa Bay, Florida | NA | NA | 0.7 ± 0.5 | Radabaugh et al. (2018) |

[a] Fluxes calculated from measurements of oxygen (O$_2$) assuming a molar CO$_2$:O$_2$ ratio of 1:1

[b] Values from figures were estimated using WebPlotDigitizer (https://automeris.io/WebPlotDigitizer/)

[c] Organic carbon value = Organic matter/1.724

[d] Organic carbon value = 95% of total carbon value



ᵉ Study did not clarify if $CO_2$ flux was measured in a light or dark chamber


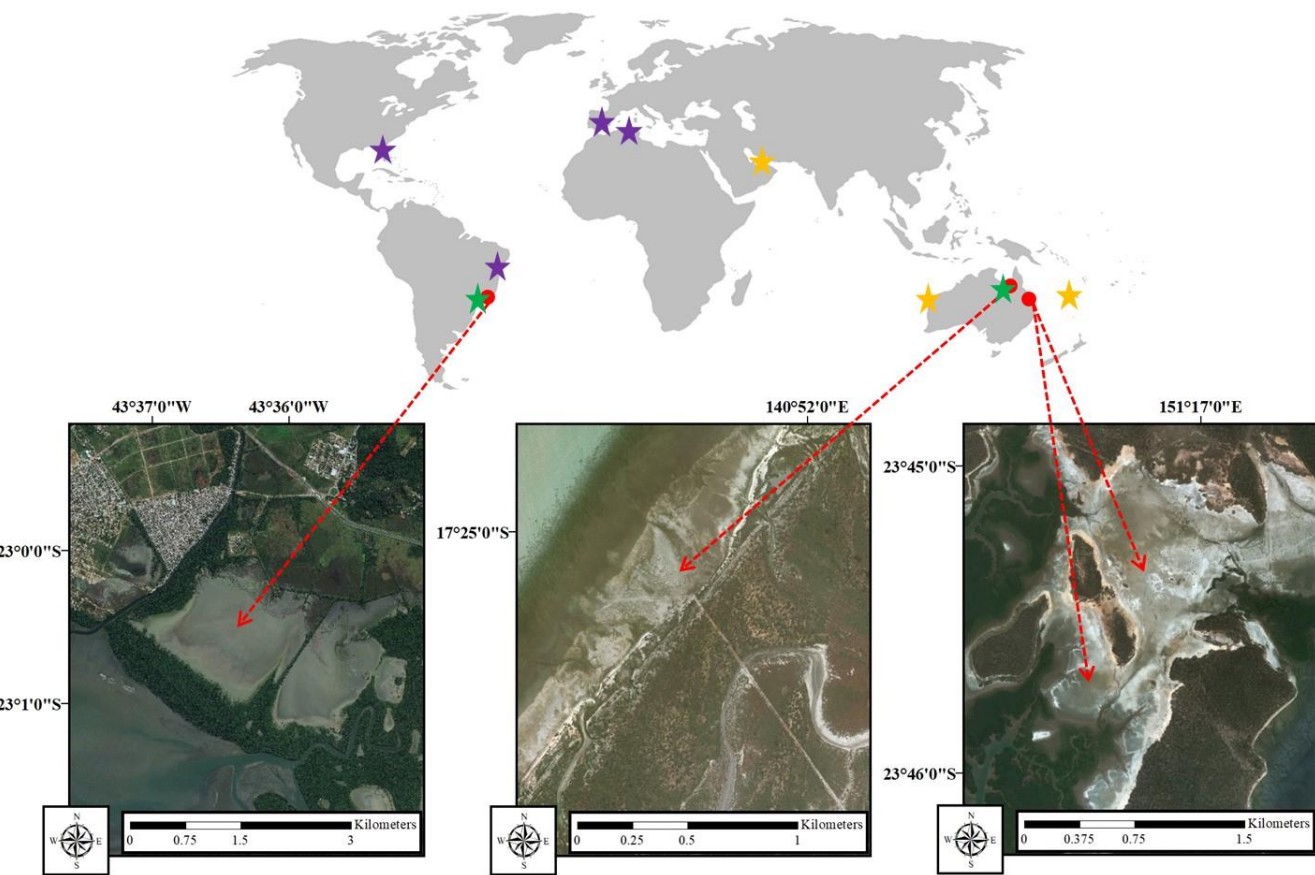

**Figure 1. Study sites (red arrows); Guaratiba, Brazil (left), Karumba, Australia (center), and Gladstone, Australia (right). Stars show the location of literature data summarized in Table 3. Purple stars are areas with only sediment carbon content data, green stars are areas with only gas flux data, and yellow stars are areas with both sediment carbon content and gas flux data from hypersaline tidal flat (HTF) studies (satellite images were taken from ©Google Earth).**



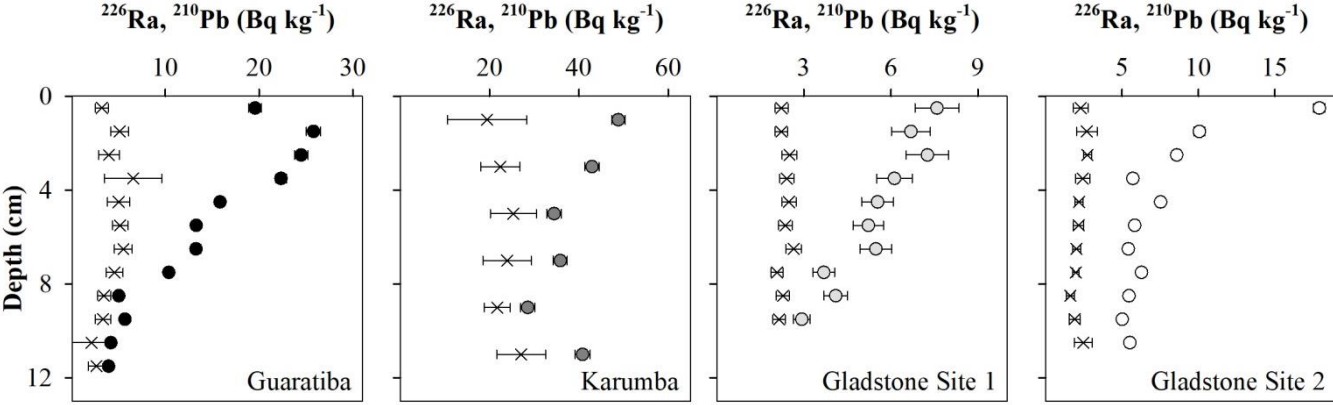


**Figure 2.** $^{226}$Ra (x) and $^{210}$Pb (circles) depth profiles of the four hypersaline tidal flat sediment cores in this work. Error bars indicate counting uncertainties.

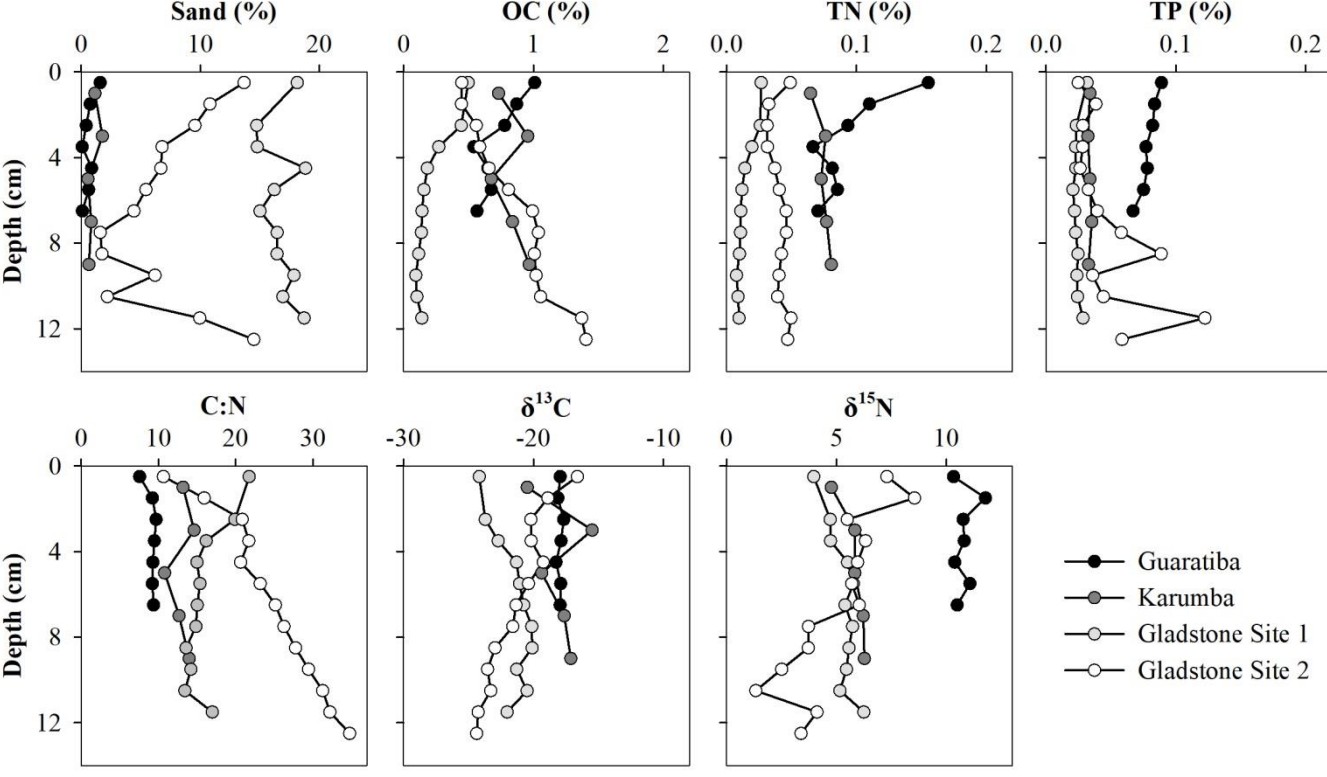

**Figure 3.** Vertical distribution of sand (>63 µm), organic carbon (OC), total nitrogen (TN) and total phosphorous (TP) contents (%) as well as $\delta^{13}$C, $\delta^{15}$N and molar C:N ratios of the four hypersaline tidal flat sediment cores.



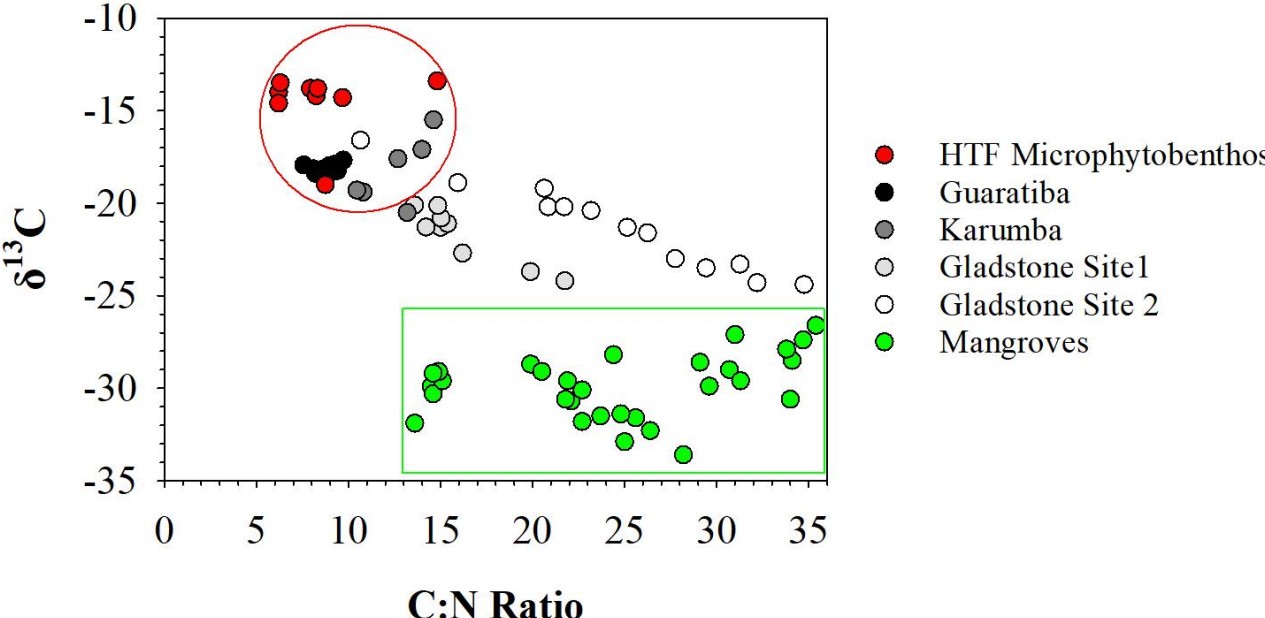

**Figure 4. Distribution of δ¹³C vs C:N molar ratio in the four hypersaline tidal flat sediment cores. Endmember values were taken from HTF surface microphytphenthos and nearby mangrove vegetation.**


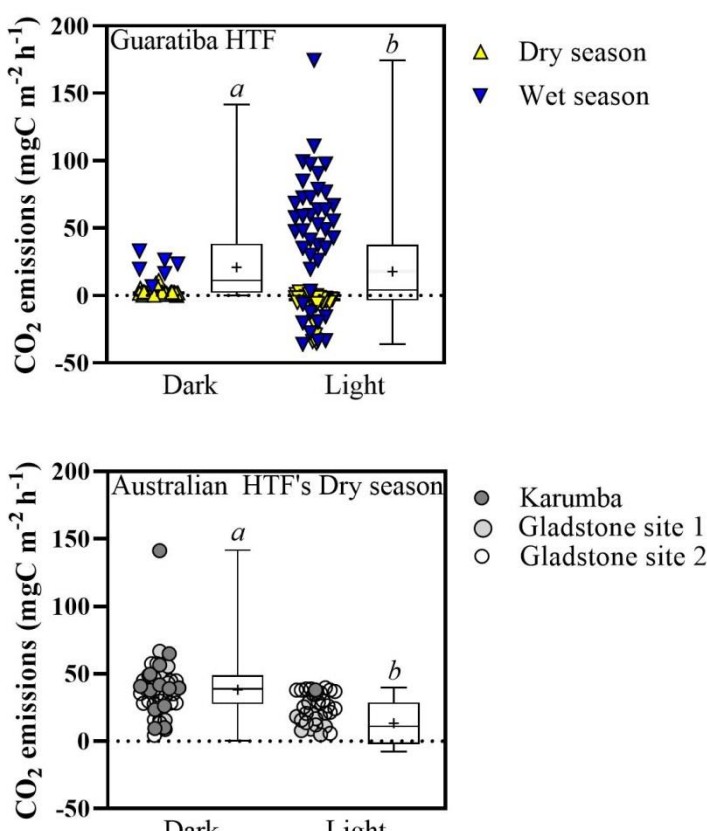

**Figure 5. Median air-sediment CO₂ fluxes (mg C m⁻² h⁻¹) from hypersaline tidal flat sediments of Guaritiba, Brazil (Dry season represented yellow triangles, n = 44 to dark chambers and n = 53 to light chambers, and wet season represented by blue triangles, n = 6 to dark chambers and n = 41 to light chambers) and Australia (Gladstone site represented by light grey circles, n = 33 to dark chambers and n = 31 to light chambers, and Karumba site represented by dark grey circles, n = 13 to dark chambers and n = 1 to light chambers). Negative values represent net CO₂ influx to sediments while positive represents net CO₂ emission to the atmosphere. Average values represented by crosses and error bars denotes whiskers minimum to maximum. Different letters indicate significant differences (Mann Whitney; p<0.05).**