# Peer review of "Hypersaline tidal flats as important "Blue Carbon" systems: A case study from three ecosystems"

_Biogeosciences, 2020_

## Referee Comment (RC1) · Anonymous Referee #1 · 2 Jan 2021

This manuscript presented an important contribution of Blue carbon from the hypersaline tidal flats widely distributed in the arid and semi-arid regions near mangrove supratidal zones, caused by microphytobenthos. The experiments are well designed and documented. I suggest acceptance of the manuscript after minor revision.

Major comments: Microphytobenthos also exist in the arid or semi-arid area near salt-marshes, as well as in the lower intertidal flats inundated daily, with much larger area than the hypersaline tidal flats in this study. Thus they also contribute to the blue car-

bon. This can be discussed as a potential for further studies as well.

The number of sites in this study is only 3 (2 in australia and 1 in brazil). Thus extrapolation of the result to the global scale still needs more studies, in addition to the estimation of hypersaline tidal flats area. This can be discussed as well.

Minor corrections: Page 2, line 65: "these microphytobenthos and are ...", remove "and". Page 3, line 77: "extent" should be "extend". Page 4, line 137: "A total 6..." should be "a total of 6..." Page 9, line 297: delete "in" at the end of this line. Page 10, line 326: "polices" should be "policies".

---

## Referee Comment (RC2) · Begy Robert-Csaba (Referee) · 11 Jan 2021

Dear Editor,

I consider the manuscript "Hypersaline tidal flats as important "Blue Carbon" systems: A case study from three ecosystems" to be relevant in the actual global environmental context. The study is addressing important issues and bringing new perspectives, that could be further considered in the approach and management of hypersaline tidal flats (HTFs). The HTFs potential capacity of long-term $CO_2$ storage and their inclusion in the blue carbon framework should be further addressed, as these ecosystems can have a potential beneficial effect in the reduction of greenhouse gases (GHG) from the atmosphere.

[Figure]

At line 139-140, "were measured in a high-purity germanium (HPGe) planar or well gamma detectors", the use of the word "or" creates confusion whether the author refers to two different detectors or if is giving an alternative name for the same apparatus, this should be clarified for a better understanding.

At line 145, "detector efficiency determined from standard calibration" needs further explanation concerning the calibration of the detector, especially it is indicated to describe the used calibration standards ( IAEA – name and type) or, if the calibration was made by Monte Carlo modelling, the software name that was used. On the other hand, the sediment dating models (CIC), that was used in this study, was proved to be, in many cases, an idealistic model, because it assumes constant sediment deposition (constant sedimentation). In this case, considering that the surface is occasionally flooded, the sedimentation rate could vary, especially in Guaratiba. In this point, the 210Pb distribution through the sediment column shows discrepancies from the theoretical exponential decrease that is expected (Figure 2). In Gladstone site 2, only CIC model can be applied, because the dating horizon is not reached, and the column does not have the full inventory, that is essential for the CRS model.

At line 165, page 5, the assumption that the pressure in the chamber is 1 atm., in my opinion, in some cases, for example, if the chamber is exposed to sunlight, the pressure inside can increase, which could influence the CO2 exhalation.

Overall, I consider that the present manuscript contains valuable scientific information, that needs to be available to the large audience.

---

## Author Comment (AC1) · 9 Feb 2021

This manuscript presented an important contribution of Blue carbon from the hypersaline tidal flats widely distributed in the arid and semi-arid regions near mangrove supratidal zones, caused by microphytobenthos. The experiments are well designed and documented. I suggest acceptance of the manuscript after minor revision. Response: We appreciate these constructive comments, below we respond to specific comments individually.

Major comments: Microphytobenthos also exist in the arid or semi-arid area near saltmarshes, as well as in the lower intertidal flats inundated daily, with much larger area than the hypersaline tidal flats in this study. Thus they also contribute to the blue car-

[Figure]

bon. This can be discussed as a potential for further studies as well. Response: We agree, and have added this topic to the Discussion, line 260: "Furthermore, micro-phytobenthos also exist in the arid or semi-arid areas near saltmarshes, as well as in the lower intertidal flats inundated daily, which are often areas greater than vegetated areas and may contribute to blue carbon burial"

The number of sites in this study is only 3 (2 in australia and 1 in brazil). Thus ex-trapolation of the result to the global scale still needs more studies, in addition to the estimation of hypersaline tidal flats area. This can be discussed as well. Response: We have the following sentence addressing this comment, Line 257: "To improve these estimates, there is clearly a need for determining carbon and nutrient accumulation rates from additional coastal HTFs and assess their areal cover in Australia, Brazil and elsewhere."

Minor corrections: Page 2, line 65: "these microphytobenthos and are ...", remove "and". Page 3, line 77: "extent" should be "extend". Page 4, line 137: "A total 6..." should be "a total of 6..." Page 9, line 297: delete "in" at the end of this line. Page 10, line 326: "polices" should be "policies". Response: Changes made as suggested.

―――――――――――――――

---

## Author Comment (AC2) · 9 Feb 2021

I consider the manuscript "Hypersaline tidal flats as important "Blue Carbon" systems: A case study from three ecosystems" to be relevant in the actual global environmental context. The study is addressing important issues and bringing new perspectives, that could be further considered in the approach and management of hypersaline tidal flats (HTFs). The HTFs potential capacity of long-term $CO_2$ storage and their inclusion in the blue carbon framework should be further addressed, as these ecosystems can have a potential beneficial effect in the reduction of greenhouse gases (GHG) from the atmosphere. Response: We appreciate these positive comments, below we respond to specific comments individually.

[Figure]

At line 139-140, "were measured in a high-purity germanium (HPGe) planar or well gamma detectors", the use of the word "or" creates confusion whether the author refers to two different detectors or if is giving an alternative name for the same apparatus, this should be clarified for a better understanding. Response: We have reworded this sentence for clarity, Line 142; "Radionuclides from the uranium-238 (238U) decay series were measured in a high-purity germanium (HPGe) gamma detectors, a planar for the Gladstone and Guaratiba and a well detector for the Karumba samples."

At line 145, "detector efficiency determined from standard calibration" needs further explanation concerning the calibration of the detector, especially it is indicated to describe the used calibration standards ( IAEA – name and type) or, if the calibration was made by Monte Carlo modelling, the software name that was used. On the other hand, the sediment dating models (CIC), that was used in this study, was proved to be, in many cases, an idealistic model, because it assumes constant sediment deposition (constant sedimentation). In this case, considering that the surface is occasionally flooded, the sedimentation rate could vary, especially in Guaratiba. In this point, the 210Pb distribution through the sediment column shows discrepancies from the theoretical exponential decrease that is expected (Figure 2). In Gladstone site 2, only CIC model can be applied, because the dating horizon is not reached, and the column does not have the full inventory, that is essential for the CRS model. Response: We have reworded this paragraph for clarity, Line 147; "…were calculated by multiplying the counts per minute by a correction factor that includes the gamma-ray intensity and detector efficiency determined from NIST Rocky Flat soils reference material." Also, we agree that the CRS model was not appropriate for this study, as such we opted for using the CIC model. Given the sedimentary excess Pb-210 profiles, we feel that the CIC was our best option.

At line 165, page 5, the assumption that the pressure in the chamber is 1 atm., in my opinion, in some cases, for example, if the chamber is exposed to sunlight, the pressure inside can increase, which could influence the CO2 exhalation. Response:

We now clarify this issue, as large changes in temperature are not expected over a short time interval of only 5-15 min. The text now read, Line 161: "Each short-term incubation lasted 5-15 min to achieve a linear change in CO2 concentration within the chambers, and was associated with a maximum increased temperature $\sim 2°$C in relation to external conditions, indicating no bias due to warming and subsequent changes in the inner pressure and biological activity."

Overall, I consider that the present manuscript contains valuable scientific information, that needs to be available to the large audience. Response: Once again, we that the reviewers for these reviews.